# An AI-Enabled All-In-One Visual, Proximity, and Tactile Perception Multimodal Sensor

Menghao Pu, Tiyong Zhao, Lingxi Zhang, Chaoqun Han, Zhiping Chai, Yifan Zhou, Han Ding, *Senior Member, IEEE* and Zhigang Wu, *Member*

*Abstract*—Visual, proximity, and tactile perception are essential sensing modalities for providing comprehensive information in interactive robotic tasks. However, integrating multiple sensors poses several challenges, including increased volume and cost, difficulties with signal synchronization and multi-sensor cross-interference or signal disruption. To tackle these challenges, we propose the vision-proximity-tactility sensor (VPTS), an AI-enabled, all-in-one multimodal sensor designed for holistic perception through efficient collaboration and information transfer between modalities, enabling complex, long-sequence robotic interactions. Facilitated by a transparent membrane patterned with ultraviolet (UV)-excited fluorescent markers, VPTS utilizes a focus-adjustable monocular camera to switch between visual, proximity, and tactile perception modalities in a time-division mode. It switches modalities by toggling UV light, camera focus, and three corresponding dedicated deep learning models. VPTS achieves an F1 score of 0.9733 in visual perception, 5.098 mm mean absolute error in proximity estimation, and 0.653 mN root–mean square error in force sensing. Real-world experiments, such as a computer music game involving up to 28 consecutive subtasks, show a cohesive pipeline where different sensing modalities collaboratively support such long-sequence manipulations, verifying VPTS's effectiveness for intricate, multimodal interactive tasks.

*Keywords—all-in-one multimodal sensor, deep learning, force sensing, proximity estimation, visual perception*

## I. INTRODUCTION

Interactive robotic tasks require accurate perception across a wide range of sensory modalities, including vision, proximity, and tactile sensing [1-7]. These modalities provide essential information for precise manipulations [8], [9], such as object localization, safe motion planning, and contact force estimation. Current systems typically rely on separate sensors for each modality, resulting in increased volume and cost [10], difficulties with signal synchronization and alignment [11], and multi-sensor cross-interference or signal disruption [12], [13].

In this work, we propose VPTS (Vision-Proximity-Tactility Sensor), an all-in-one multimodal sensor that integrates visual, proximity, and tactile modalities into a compact, cost-effective design. VPTS employs a transparent membrane together with a

This work is partially supported by the National Natural Science Foundation of China Under Grant (52188102), National Key Research and Development Program of China Under Grant (2024YFB4707902), Cross-research Support Program of Huazhong University of Science and Technology (2024JCYJ036).

Menghao Pu, Tiyong Zhao, Lingxi Zhang, Chaoqun Han, Zhiping Chai, Yifan Zhou, Han Ding, Zhigang Wu are with the school of mechanical science and engineering, Huazhong University of Science and Technology, Wuhan, China. Corresponding author e-mail: zgwu@hust.edu.cn.

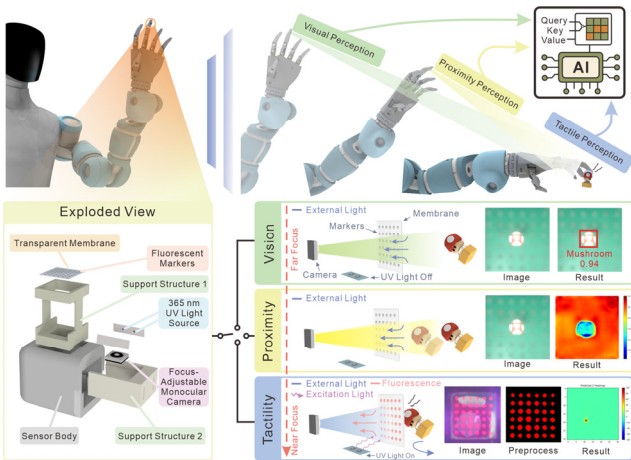

Figure 1. AI-enabled All-in-one Multimodal VPTS for visual, proximity, and tactile perception.

focus-adjustable camera, allowing environmental perception when the ultraviolet (UV) light is off and the camera is set to far-focus, and deformation sensing of fluorescent markers when the UV light is on and the camera is set to near-focus. Each modality is subsequently processed by a dedicated deep learning module, enabling a holistic understanding of the interaction scenario through object recognition, depth estimation, and force generation, Figure 1. VPTS offers a novel solution that overcomes the limitations of conventional systems and enables more complex, long-sequence multimodal perception interactive robotic tasks.

## II. SENSOR DESIGN AND ALGORITHM PIPELINE

VPTS is primarily composed of a transparent membrane, UV-excited fluorescent markers, a focus-adjustable monocular camera, a 365 nm UV light source, Figure 1. With the UV light off and the camera in far focus, VPTS captures external scenes for visual and proximity sensing. When the UV light is on and the camera switches to near focus, fluorescence from the markers encodes membrane deformations for tactile sensing. This compact design unifies multiple modalities in a single unit, reducing hardware redundancy and enabling smooth transitions.

VPTS utilizes a deep learning-based algorithmic pipeline that seamlessly integrates visual, proximity, and tactile perception modules, Figure 2. For the visual perception module, we adopt a transformer-based architecture, whose attention mechanism can capture both global and local features and adapt to visually disturbed or noisy environments [14], [15]. For the proximity perception module, we adopt a hybrid strategy that combines classification and depth-map regression [16]. For the

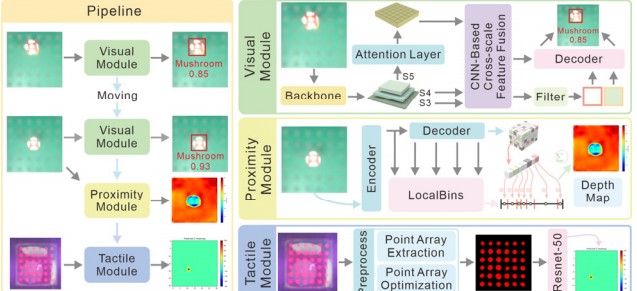

Figure 2. Utilization of deep learning pipeline integrating three perception modalities.

tactile perception module, we employ a combination of image preprocessing and ResNet-50 to generate detailed force distribution maps.

## III. RESULTS AND APPLICATIONS

The visual perception module demonstrates high accuracy and robustness in both single and multiple object detection scenarios, achieving an F1 score of 0.9733 and effectively handling variations in object position and visual conditions. The proximity perception module enables precise monocular depth estimation with a mean absolute error of 5.098 mm, showing strong stability and generalizability across different distances. The tactile perception module exhibits excellent force estimation accuracy with a RMSE of 0.653 mN and a Pearson correlation of 0.9967, confirming its reliability in capturing distributed force for robotic manipulation, Figure 3.

We also conduct an interactive music game, where VPTS plays the song "Twinkle Twinkle Little Star" on a keyboard based on the position of a black tile in the game, Figure 4. First, VPTS faces the screen and uses the vision module to identify the target key. Then it turns to the keyboard and locates the key position. After that, the proximity module estimates the distance. The robotic arm moves accordingly. Finally, the tactile module monitors the distributed force. Once the force threshold is reached, the system releases the key to finish. The system successfully completes up to 28 perception and action sub-tasks, Table 1, demonstrating its ability to perform intricate, long-sequence tasks that require continuous sensory feedback.

## IV. CONCLUSION AND FUTURE WORK

VPTS is a powerful, compact multimodal sensor that combines visual, proximity, and tactile perception for robotic tasks. It achieves high accuracy across all modalities and demonstrates strong real-world applicability in tasks requiring continuous sensory feedback. Future work will focus on optimizing system size and enhancing model generalization.

Table 1 Comparison of sequential perception capability.

| Ref. | | Total time (s) | Subtask count | Average time (s) | Modality count |
|---|---|---|---|---|---|
| Ref [17] | Task 1 | 8 | 1 | 8 | 1 |
| Ref [18] | Task 1 | 180 | 4 | 45 | 3 |
| | Task 2 | 105 | 4 | 26.25 | 3 |
| Ref [19] | Task 1 | 33 | 1 | 33 | 2 |
| **VPTS (ours)** | Task 1 | 26.5 | 3 | 8.83 | 3 |
| | Task 2 | 25.5 | 3 | 8.5 | 3 |
| | Task 3 | 372 | **28** | 13.3 | 3 |

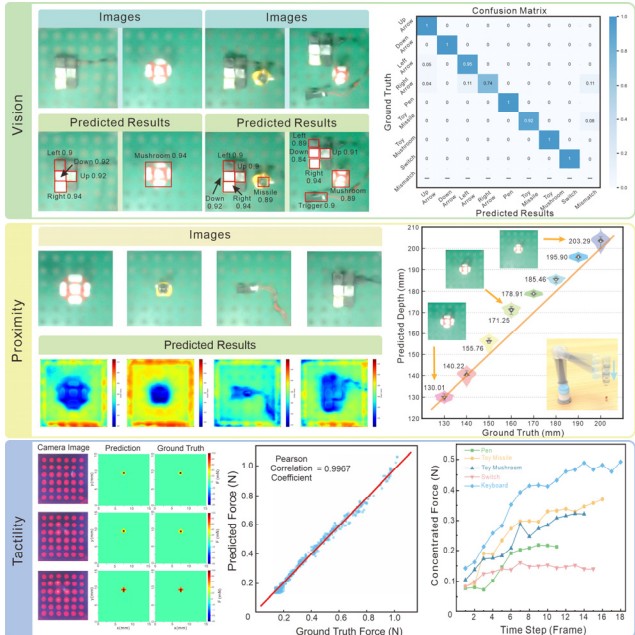

Figure 3. Demonstration of VPTS's visual, proximity, and tactile perception capacity.

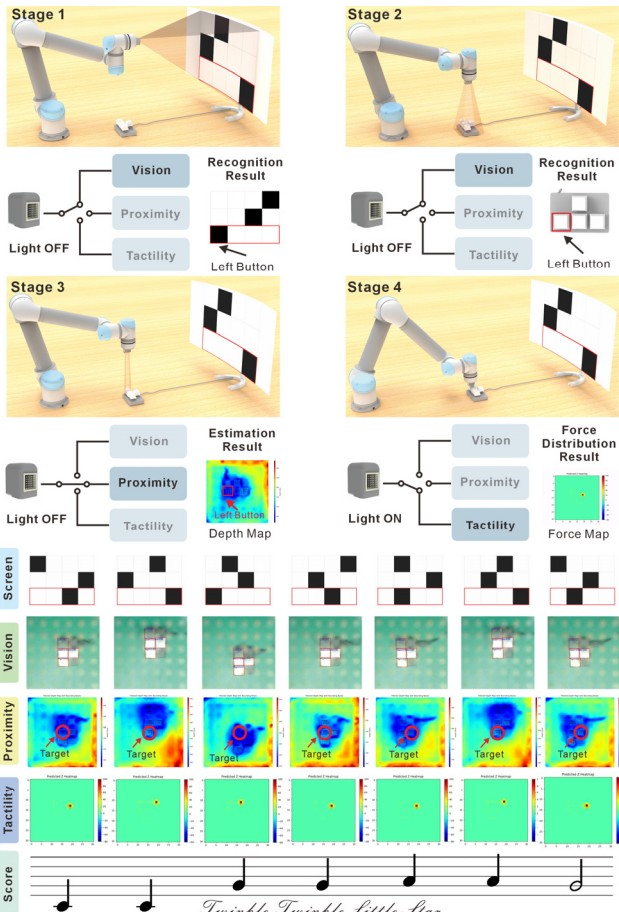

Figure 4. A robotic arm equipped with VPTS playing "*Twinkle Twinkle Little Star*".

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
