# OpenReview forum: "An AI-Enabled All-In-One Visual, Proximity, and Tactile Perception Multimodal Sensor"
_IEEE.org/IROS/2025/Workshop/Tactile_Sensing — IROS 2025 Workshop Tactile Sensing Poster_

### Official Review · Reviewer_w3an · 2025-09-19
**Review of An AI-Enabled All-In-One Visual, Proximity, and Tactile Perception Multimodal Sensor**

**Rating:** 7
**Confidence:** 4

**Review:**

The paper presents an all-in-one visual, proximity, and tactile perception multimodal sensor. The presentation is well-made and the demonstration is interesting. Some suggestions for improvements are as follows.
1. The authors should clarify how each sensing modality is activated during operation.
2. The setup for the demonstration of using the keyboard should be explained in more detail.
3. Although the figures contain a great deal of information, the accompanying text does not convey all of it, making the paper hard to follow. Given the space constraints, I suggest rearranging or consolidating the figures to free up room for more explanatory text.

---

### Official Review · Reviewer_6mhW · 2025-09-22
**A multimodal sensor for long-sequence interactive robotic tasks**

**Rating:** 6
**Confidence:** 4

**Review:**

This paper presents a multimodal sensor for long-sequence interactive robotic tasks, combining UV-light switching with deep learning algorithms to achieve object recognition, proximity estimation, and force sensing. The integration of different modalities into a compact system is interesting and shows potential for real-world applications. That said, I feel the paper could be strengthened in a few areas:

(1) The description of the deep learning modules and their prediction targets could be clarified further to help readers better understand the design choices.

(2) Since the monocular depth-based proximity estimation is critical for long-sequence tasks, it would be valuable to include experiments or analyses that demonstrate its accuracy and reliability.

(3) The presentation of the experimental demonstrations could be made clearer. The current use of musical staff and blocks may be somewhat difficult for readers to follow, and a more intuitive setup—such as a piano keyboard demonstration—might help convey the idea more effectively.

Overall, this is a promising piece of work. With additional clarification and refinement of the experiments, the paper would be even more convincing and impactful.